# How Useful Is TikTok for Patients Searching for Carpal Tunnel Syndrome-Related Treatment Exercises?

**DOI:** 10.3390/healthcare12171697

**Published:** 2024-08-26

**Authors:** Damon V. Briggs, Albert T. Anastasio, Mikhail A. Bethell, Joshua R. Taylor, Marc J. Richard, Christopher S. Klifto

**Affiliations:** 1School of Medicine, Duke University, Durham, NC 27710, USA; mikhail.bethell@duke.edu; 2Department of Orthopaedic Surgery, Duke University Medical Center, Durham, NC 27710, USA; albert.anastasio@duke.edu (A.T.A.); marc.richard@duke.edu (M.J.R.); christopher.klifto@duke.edu (C.S.K.); 3School of Medicine, University of North Carolina, Chapel Hill, NC 27599, USA; joshua.taylor@duke.edu

**Keywords:** TikTok, carpal tunnel, educational value, DISCERN, social media

## Abstract

Since orthopedic surgery has been slower to acknowledge the rise of social media for distributing medical information, this study aims to evaluate TikTok videos’ quality and educational value in relation to carpal tunnel syndrome treatment exercises. TikTok was searched using the hashtags “#carpaltunnelexercises”, “#carpaltunnelremedies”, “#carpaltunnelrehab”, and “#physicaltherapyforcarpaltunnel”. The engagement indicators were documented and the video content quality was assessed using the DISCERN, CTEES, JAMA, and GQS grading scales. There were 101 videos included, which accumulated 20,985,730 views. The videos received 1,460,953 likes, 15,723 comments, 243,245 favorites, and 159,923 shares. Healthcare professionals were responsible for 72% of the video uploads, whereas general users contributed 28%. More healthcare professionals’ videos were graded as “poor” (79%) compared to general users (21%). General users received slightly more video grades of “very poor” (52%) than healthcare professionals (48%). For the DISCERN grading, the videos by healthcare professionals were significantly better than those by general users in terms of reliability, achieving aims, and relevancy. They were also superior in the overall composition of the health information derived from the total DISCERN score. However, no significant differences were found between the two groups when using the CTEES, JAMA, and GQS grading scales. Overall, despite the emergence of TikTok as a medical information tool, the quality and educational value of the carpal tunnel syndrome exercise videos were poor.

## 1. Introduction

The growth of health-related content innovation on TikTok is concerning due to the platform’s broad outreach and the absence of review processes that are similar to those of academic journals and news organizations. More than 67% of Americans use social media as a news source [1]. As a result, patients access health information and services without presenting to healthcare providers, which increases the self-management of conditions [2]. However, the dissemination of misinformation through social media has been a major problem, and the COVID-19 pandemic has re-emphasized the consequences of misinformation for public health [1]. Amongst all the social media platforms, TikTok has a powerful influence because it continues to grow exponentially and has widespread popularity. Furthermore, its appeal extends beyond younger generations, as it has been adopted among older individuals [3,4]. In orthopedics, sharing inaccurate or insufficient health information on social media, specifically TikTok, can lead to adverse effects for patients and viewers, such as exercising improperly, worsening injuries, and creating uncertainty for proper rehabilitation strategies and treatments. In fact, the variability and low-quality of YouTube videos on femoroacetabular impingement, patellar dislocations, and ulnar collateral ligament injuries highlight how the information on social media may be low-quality, deliver inadequate health information, and be predisposed to bias [5,6,7,8]. Therefore, assessing the quality, dependability, and correspondence with the current literature of health information on TikTok is key to protecting the health of individuals who trust this material to make informed decisions [9,10].

Carpal tunnel syndrome (CTS), the predominant compressive neuropathy affecting the upper extremity, is described as numbness, pain, or tingling in the hands and wrists [11]. Its incidence ranges from 1 to 3 patients per 1000, with a more significant occurrence amongst women. Typically, the affected age demographic range is 45 to 60 years old [12]. Non-surgical management is common and includes wrist splints and non-steroidal anti-inflammatory drugs (NSAIDs), which provide immobilization and support and reduce the discomfort and inflammation, respectively [11,13]. Although surgery is considered definitive treatment, physical therapy exercises for wrist and hand strengthening, stretching, and nerve glides are moderately effective in short- and mid-term CTS treatment [14].

While many studies have performed quality assessments of the health information available on TikTok across different areas, such as COVID-19 [15], back pain [16], and neurosurgery [17], more research is needed regarding orthopedic surgery topics on this platform. Thus, recognizing TikTok as a medical education tool coupled with the goal of entertainment, this study aimed to analyze TikTok videos’ educational value, quality, and reliability in relation to CTS-related exercises. We hypothesized that the content of these videos would be of poor quality, as determined via low scores on different grading scales, and the number of videos by general TikTok users would exceed those by qualified healthcare professionals.

## 2. Materials and Methods

### 2.1. Search Strategy and Data Collection

The social media platform, TikTok, was queried to find videos related to carpal tunnel exercises on 24 April 2023. The search used the hashtags “#carpaltunnelexercises”, “#carpaltunnelremedies”, “#carpaltunnelrehab”, and “#physicaltherapyforcarpaltunnel” without any search filters. Thus, through our search terms, we intended to capture most of the videos that a TikTok user would most likely encounter when searching the platform for CTS-related exercises. The search yielded many videos (n = 370). We then performed an initial screening of the videos to eliminate those that were (1) duplicates (n = 135), (2) not related to CTS (n = 20), (3) not related to carpal tunnel exercises (n = 74), (4) not in English (n = 6), (5) pertaining to hand taping (n = 32), and (6) unavailable (n = 3). After the screening, 101 videos remained for the data analysis (Figure 1).

We documented the data of each video analyzed, which consisted of the creator’s information (username and video) and the number of views, likes, shares, comments, and favorites. Since this study did not involve human participants or animals, there was no need to seek approval from an ethics committee.

### 2.2. Scoring System

We employed four distinct scoring systems to assess the videos’ quality and educational value. The DISCERN tool, a validated method for evaluating treatment reliability and quality, was utilized. Additionally, the carpal tunnel exercise education score (CTEES) was employed to assess the suitability of the educational information in each video. The Global Quality Scale (GQS), which utilizes a five-point grading system, was used to determine the video quality. Finally, the *Journal of American Medical Association* (JAMA) criteria were employed to evaluate the quality and reliability of the videos. The CTEES score was adapted from a similar scale developed by Jang et al. [18] for assessing scoliosis exercise video quality in previous research.

### 2.3. DISCERN for Reliability and Quality Assessment

The DISCERN questionnaire is a reliable and precise tool that researchers utilize to evaluate the quality of information regarding treatment options for health issues. This well-validated tool has been in use since the late 1990s and consists of 16 questions [19]. The initial eight questions (DISCERN 1) assess the publication’s reliability. The subsequent set of seven questions (DISCERN 2) examine the author’s source base in terms of the quality. The final question (DISCERN 3) rates the publication as a source of information. Although initially designed for written materials, DISCERN has been successfully adapted as a scoring test to assess the quality of videos in previous research [20]. The DISCERN scores are categorized as follows: scores ranging from 63 to 75 points are considered excellent, 51 to 62 points are regarded as good, 39 to 50 points as fair, 27 to 38 points as poor, and 16 to 26 points as very poor.

### 2.4. CTEES for Educational Suitability Assessment

In order to assess the educational value of the videos, we introduced the CTEES, which is a modified version of a scale developed by Jang et al. [18] This scoring system evaluates whether viewers can understand and follow the exercises demonstrated in the video. The CTEES comprises five grading criteria: “Exercise cycle” (description of the exercise sequence), “Target” (explanation of the targeted area for the exercise), “Effect” (description of the expected impact of the exercise), “Safety” (inclusion of precautions and safety aspects), and “Rationale” (explanation of the exercise’s underlying rationale). Each criterion is assigned a score ranging from 0 to 5, with higher scores indicating higher quality. The cumulative score for all five criteria yields the final CTEES, ranging from 0 to 25. A score of 0 represents the lowest possible quality, while 25 represents the highest possible quality.

### 2.5. Global Quality Scale (GQS) for Assessing the Overall Quality

The GQS was employed to assess the overall quality of the videos, aiding in distinguishing between videos of low and high quality. The GQS utilizes a five-point grading scale, where scores ranging from 1 to 2 indicate low-quality videos, a score of 3 indicates medium quality, and scores of 4 to 5 are associated with high-quality videos [21].

### 2.6. Journal of American Medical Association (JAMA)

The JAMA benchmark criteria employ a four-point scale to evaluate the quality and reliability of medical information. This scale involves assessing four key aspects: (1) the credentials and affiliations of the authors, (2) the presence of sources and references, (3) the disclosure of information regarding sponsorships and endorsements, and (4) the dates when the content was posted. JAMA scores of 0–1, 2–3, and 4 represent insufficient, partially sufficient, and completely sufficient information, respectively [22,23].

### 2.7. Video Assessment

The videos were collected by two authors and independently evaluated by our orthopedic research team. Once the data regarding video distribution metrics were collected for each video, the content of the videos was graded using the DISCERN, CTEES, JAMA, and GQS tools. Each video was graded separately by two trained reviewers. A third author resolved any points of discrepancy between the two reviewers.

After scoring the videos, they were categorized into groups based on the uploader’s credentials: general users, healthcare providers, and health organizations. The healthcare provider category included users that provided their credentials in their TikTok name or description, such as chiropractors, physicians, physical therapists, and nurses. The health organization group includes clinics, hospitals, and treatment centers. It is important to note that the reviewers were not blinded to whether healthcare professionals or general users uploaded the videos during the video-grading process.

### 2.8. Statistical Analysis

The scoring and characteristic data are presented as the mean (standard deviation) (SD), median (inter-quartile range [IQR]), and percentage. A two-sample *t*-test was used to compare the two types of uploaders by utilizing each continuous and categorical variable’s mean, standard deviation, and sample size. The interobserver reliability of the DISCERN, CTEES, JAMA, and GQS grading scales was determined using the class 3 model intraclass correlation coefficient (ICC). A 2-factor analysis of variation without replication model was employed to calculate the ICC, along with a 95% confidence interval. The benchmarking of the ICC values was based on previous research, with the following classification: less than 0.50 indicating poor reliability, 0.50 to 0.75 indicating moderate reliability, 0.76 to 0.90 indicating good reliability, and values exceeding 0.90 indicating excellent reliability [24]. Statistical significance was set at *p* < 0.05 for comparisons other than interobserver reliability. All the analyses were performed using Microsoft Excel^®^ version 16.88 (Redmond, WA, USA).

## 3. Results

### 3.1. Basic Characteristics

The basic features of the 101 examined videos are displayed in Table 1. The 101 videos received 20,985,730 views, with a median of 17,800 (IQR = 4371 to 70,700). Additionally, the videos obtained 1,460,953 likes, 15,723 comments, 243,245 favorites, and 159,923 shares, with a median of 735 (IQR = 150 to 2810), 14 (IQR = 2 to 51), 135 (IQR = 38 to 605), and 88 (IQR = 18 to 482), respectively.

### 3.2. Types of Uploaders

Healthcare professionals uploaded the majority of the videos (n = 73, 72.3%) when compared to general users (n = 28, 27.7%) (Table 1). Among the videos shared by healthcare professionals, the breakdown was as follows: 25 by a doctor of physical therapy; 19 by chiropractors; 6 by physiotherapists; 4 by physical therapists with either bachelor’s or master’s degrees; 4 by Chicago Sports and Chiro, a chiropractic group, 3 by occupational therapists; 3 by Hulst Jepsen PT, a physical therapy group; 1 by a manual osteopathic practitioner; 1 by an acupuncturist; and 1 by each of the following healthcare practices/groups: IRG PT & OT, Magno Physical Therapy, Shreveport, LA PT clinic, Precision Care Medical (PT & OT), and Path Medical (PT & Chiropractic care).

### 3.3. DISCERN Scores

The higher DISCERN 1 and total DISCERN scores for the videos by healthcare professionals were statistically significant (*p* < 0.001 and *p* = 0.013, respectively) when compared to general users (Table 2). Regarding the separate domains of DISCERN 1, the videos by general users scored significantly lower in reliability, achieving aims, and relevancy (*p* < 0.001). Overall, the DISCERN 1, DISCERN 2, DISCERN 3, and total DISCERN scores, with their associated breakdowns, were higher for videos uploaded by healthcare professionals than for general users. However, all the values were less than 50% of the maximum respective scoring, and the DISCERN 2 and 3 scores were statistically insignificant.

### 3.4. CTEES

In regard to the CTEES, general users scored slightly higher than healthcare professionals (6.43 and 5.99, respectively) on a 25-point scale (*p* = 0.650) (Table 3). However, this distinction was statistically insignificant. Ultimately, general users and healthcare professionals had very low CTEESs and associated category breakdowns, scoring less than 30% of the maximum. 

### 3.5. JAMA and GQS Scores

Similar to the very low CTEESs for videos by general users and healthcare professionals, not only were the JAMA and GQS score differences between general users and healthcare professionals statistically insignificant (*p* = 0.234 and *p* = 0.288, respectively), the scoring trend was low (Table 4). The GQS scores for both groups were categorized as low quality, with values of 2.78 and 2.63 for healthcare professionals and general users, respectively.

### 3.6. Total DISCERN Grading

The DISCERN grading results are shown in Figure 2. General users had slightly more videos graded as very poor (n = 11, 52%) than healthcare professionals (n = 10, 48%). More videos uploaded by healthcare professionals were graded as poor (n = 59, 79%) than general users (n = 16, 21%). General users had fewer videos graded as fair (n = 1, 20%) than healthcare professionals (n = 4, 80%). However, no videos between the two groups were graded as good or excellent.

### 3.7. Interobserver Reliability

The interobserver reliability approximations for the DISCERN, CTEES, JAMA, and GQS grading scales were 0.95 (95% CI, 0.93–0.97), 0.99 (95% CI, 0.98–0.99), 0.91 (95% CI, 0.86–0.94), and 0.84 (95% CI, 0.77–0.89), respectively. These DISCERN, CTEES, and JAMA findings correlate with “excellent” reliability, while those for the GQS correspond to “good” reliability.

## 4. Discussion

The findings of this study showed significant differences between TikTok videos uploaded by healthcare professionals and general users regarding the reliability, goal achievement, relevancy, and composition of health information based on the total DISCERN score. TikTok videos by healthcare professionals received significantly higher scores in the aspects mentioned above when compared to general users. However, no significant differences were found between the two groups when using the CTEES, JAMA, and GQS grading scales. The CTEES were less than 30% of the maximum value, correlating with poor educational suitability. Additionally, although most TikTok videos were from qualified healthcare professionals, more were categorized as poor quality rather than very poor quality compared to those by general users.

Over 75% of patients search the Internet their disease processes, physicians, and healthcare institutions before presenting as new patients [16,25]. Through these searches, TikTok videos are most likely included in the populating algorithm. Despite the increased use of TikTok, more research is needed on the spread of orthopedic-related information on this platform. Therefore, our objective was to assess TikTok videos’ quality and educational value regarding CTS treatment exercises because of the high video inventory addressing this pathological condition, which can affect many individuals [26,27]. This study verifies the vast utilization of TikTok for accessing CTS-related treatment exercises. The 101 videos received over 20.9 million views, with a median of 17,800 views per video, and this count continues to grow daily.

### 4.1. DISCERN, JAMA, and GQS Grading

In the data analyses, TikTok’s CTS-related exercise videos showed deficits in quality and reliability. No video received a grading higher than “fair” based on the DISCERN scoring system. Although the videos uploaded by healthcare professionals were significantly better in terms of the reliability, aim achievement, and relevancy than those by general users (*p* < 0.001), the average total DISCERN score for all the videos was less than 50% of the maximum. The JAMA and GQS grading further validated the low video quality and reliability, with total average scores of 2.05 and 2.74, respectively. No significant differences were found between the two groups for these measures. These findings show TikTok videos’ insufficiencies in describing safety precautions, aspects of treatment exercises, and the rationale or effects of the movements. Furthermore, since TikTok lacks the means to cite sources properly, the reliability of the videos is further reduced through the JAMA scoring.

### 4.2. CTEES

The total CTEES was less than 25% of the maximum, and no significant differences were identified between the two groups. This result reveals the lack of educational value of TikTok CTS-related treatment exercise videos. As a result, viewers are unable to learn fundamental aspects, such as exercise routines, targets, effects, safety measures, and underlying treatment rationales.

### 4.3. Uploaders

Contrary to our hypothesis, qualified healthcare professionals uploaded most videos (72%). However, the videos by general users secured more views, likes, and favorites. Various factors can account for the difference in engagement characteristics between videos by healthcare professionals and general users, such as how general users tend to focus on entertainment and trends to captivate viewers and have a more relatable style of communication. Overall, the contribution from healthcare professionals did not significantly augment the video quality, educational appropriateness, and effectiveness. Therefore, healthcare professionals should engage in plans to revamp these videos and ensure the spread of precise information to patients and viewers.

### 4.4. Comparison with Previous Research

Altogether, our findings support previous research that identified TikTok’s propagation of low-quality information on various orthopedic topics [28,29,30]. For instance, a recent study by Bethell et al. investigated the quality, educational value, and distribution of videos regarding shoulder instability exercises. Using the DISCERN and SSEES scoring systems, they reported total scores of 24.6 and 3.8, respectively [29]. Congruent with our results, these findings signify that despite the videos by healthcare professionals achieving significantly higher scores for the quality, reliability, and educational value, the overall educational value of the videos was poor [29]. Similarly, Anastasio et al. evaluated TikTok’s quality and educational utility in terms of ankle sprain treatment exercises using the DISCERN and ASEES scores. They discovered that the total DISCERN and ASEES scores were 28.0 and 8.9, respectively. In agreement with our study, these results showed that none of the videos qualified for “good” or “excellent” DISCERN grades, correlating with poor educational value [30]. Plausible explanations for the poor-quality videos by healthcare professionals specifically are the tendency to share evidence-based medicine practices in terms that are less comprehendible for the everyday viewer and sacrificing accuracy or supportive explanations for entertainment in order to maintain viewer engagement on the platform. However, some studies have recognized TikTok’s value for spreading higher-quality health evidence on issues outside of orthopedics, such as diabetes [31] and obstructive lung disease [32]. Ultimately, while TikTok displays diverse material on CTS-related exercises, its reliability is limited by the absence of scientific regulation of video quality.

### 4.5. Limitations

There are several limitations concerning this study of the quality and educational value of TikTok’s CTS-related treatment exercise videos. Firstly, there is the risk of selection bias due to the specific search terms implemented. We chose “carpaltunnelexercises”, “carpaltunnelremedies”, “carpaltunnelrehab”, and “physicaltherapyforcarpaltunnel” to imitate the search behavior of interested users. However, since users knowledgeable about CTS may use other search terms, sampling bias is possible. Secondly, the video-grading processes are affected by observer bias due to the subjectivity of assessing content quality. To mitigate this bias, we used the DISCERN, JAMA, GQS, and CTEES grading scales, which are all well validated. We utilized these tools for a more objective assessment. Thirdly, since the reviewers were not blinded as to whether healthcare professionals or general users uploaded the videos during the grading process, there is a potential for observer bias. Lastly, regarding the interobserver inconsistency, each video was independently assessed by two different reviewers. A third reviewer resolved discrepancies when applicable to minimize the effect of subjectivity.

## 5. Conclusions

As predicted by the CTS-related exercise videos’ poor quality and educational value, TikTok requires improvements to be an adequate medical information tool. Videos uploaded by healthcare professionals were significantly superior to those by general users in terms of quality across multiple domains of DISCERN. Despite this significance, the educational value of videos from both groups was poor. Approximately 5% of the videos were graded as “fair”, with none receiving “good” or “excellent” ratings. Acknowledging the impact and reach of low-quality TikTok content, healthcare professionals should advise patients to approach the information presented on the platform with caution.

## Figures and Tables

**Figure 1 healthcare-12-01697-f001:**
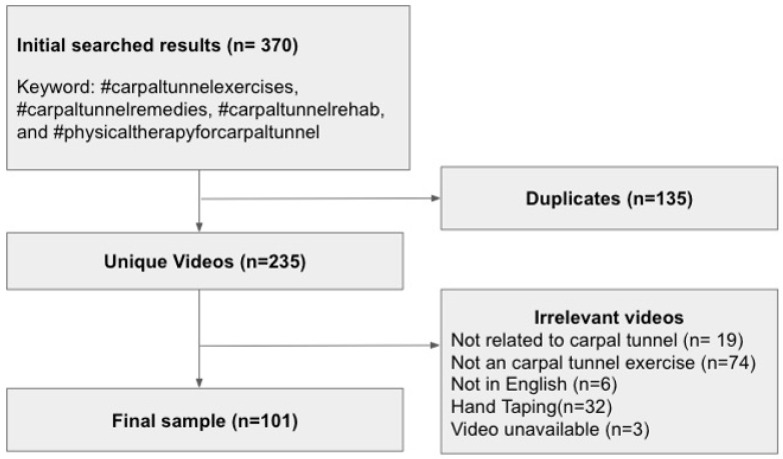
Flowchart of the search process for videos related to carpal tunnel exercises.

**Figure 2 healthcare-12-01697-f002:**
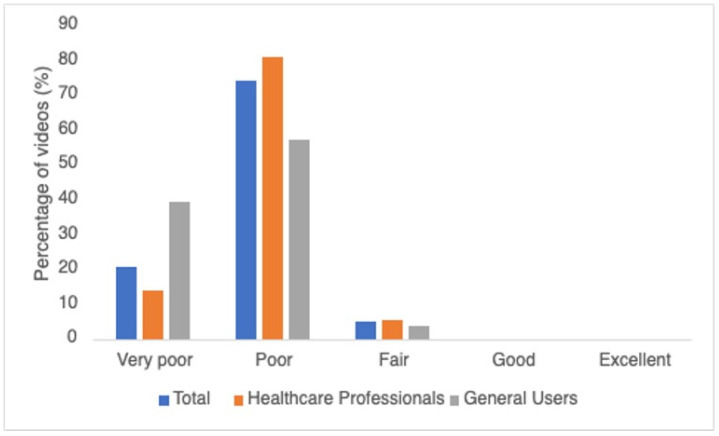
Quantity of the DISCERN grades for videos by two types of uploaders.

**Table 1 healthcare-12-01697-t001:** Video characteristics.

Scoring, Median (IQR)	Total (n = 101)	Healthcare Professionals (n = 73)	General Users (n = 28)
Number of views	17,800.0 (4371.0–70,700.0)	17,500.0 (4066.0–70,700.0)	30,250.0 (6517.5–72,125.0)
Likes	735.0 (150.0–2810.0)	724.0 (150.0–2808.0)	803.0 (184.3–3072.0)
Comments	14.0 (2.0–51.0)	17.0 (2.0–51.0)	10.5 (3.5–45.8)
Favorites	135.0 (38.0–605.0)	126.0 (38.0–605.0)	209.0 (44.0–485.0)
Shares	88.0 (18.0–482.0)	109.0 (18.0–482.0)	56.5 (18.8–467.0)

**Table 2 healthcare-12-01697-t002:** DISCERN scores. Bold *p* values represent significant figures.

Scoring, Mean (SD)	Total (n = 101)	Healthcare Professionals (n = 73)	General Users (n = 28)	*p* Value
DISCERN 1	16.28 (1.65)	17.12 (1.29)	15.04 (1.51)	**<0.001**
Reliable?	3.50 (0.75)	3.90 (0.29)	2.46 (0.57)	**<0.001**
Achieve aims?	3.02 (0.46)	3.11 (0.46)	2.79 (0.36)	**<0.001**
Relevant?	3.00 (0.30)	3.05 (0.28)	2.89 (0.34)	**<0.001**
Clear on sources of info?	1.03 (0.21)	1.04 (0.24)	1.00 (0.00)	NA
Clear on when info was published?	1.00 (0.11)	1.01 (0.13)	1.00 (0.00)	NA
Balanced and unbiased?	2.79 (0.39)	2.80 (0.39)	2.77 (0.39)	0.730
Details for additional info?	1.04 (0.25)	1.05 (0.29)	1.00 (0.00)	NA
Address areas of uncertainty?	1.15 (0.44)	1.16 (0.45)	1.13 (0.41)	0.759
DISCERN 2	11.07 (2.55)	11.08 (2.51)	11.05 (2.66)	0.958
How treatment works?	1.59 (0.95)	1.60 (0.96)	1.55 (0.91)	0.813
Benefits of each?	1.32 (0.71)	1.32 (0.71)	1.32 (0.70)	1.000
Risks of each?	1.04 (0.24)	1.05 (0.28)	1.00 (0.00)	NA
Risks of no treatment?	1.03 (0.21)	1.01 (0.08)	1.07 (0.37)	0.191
Quality of life effect?	1.06 (0.26)	1.04 (0.22)	1.11 (0.34)	0.226
More than one treatment possible?	2.49 (0.88)	2.45 (0.88)	2.59 (0.88)	0.476
Support shared decision making?	2.54 (0.74)	2.60 (0.73)	2.41 (0.73)	0.245
DISCERN 3	2.32 (0.66)	2.37 (0.66)	2.20 (0.65)	0.247
Total DISCERN	29.94 (4.21)	30.58 (4.02)	28.29 (4.24)	**0.013**

**Table 3 healthcare-12-01697-t003:** CTEES scores.

Scoring, Mean (SD)	Total (n = 101)	Healthcare Professionals (n = 73)	General Users (n = 28)	*p* Value
CTEES	6.11 (4.35)	5.99 (4.41)	6.43 (4.16)	0.650
Does it describe exercise cycle?	2.85 (0.60)	2.81 (0.67)	2.95 (0.39)	0.302
Does it describe target of cycle?	1.15 (1.38)	1.20 (1.40)	1.00 (1.33)	0.516
Does it describe effect of exercise?	1.09 (1.33)	1.05 (1.37)	1.21 (1.21)	0.589
Does it describe precautions?	0.22 (0.70)	0.19 (0.61)	0.30 (0.90)	0.482
Does it explain exercise rationale?	0.80 (1.29)	0.74 (1.29)	0.96 (1.29)	0.445

**Table 4 healthcare-12-01697-t004:** JAMA and GQS scores.

Scoring, Mean (SD)	Total (n = 101)	Healthcare Professionals (n = 73)	General Users (n = 28)	*p* Value
JAMA	2.05 (0.23)	2.03 (0.17)	2.09 (0.33)	0.234
Authorship	1.00 (0.00)	1.00 (0.00)	1.00 (0.00)	NA
Attribution	0.04 (0.20)	0.03 (0.16)	0.07 (0.26)	0.352
Disclosure	0.01 (0.07)	0.01 (0.06)	0.02 (0.09)	0.519
Currency	1.00 (0.00)	1.00 (0.00)	1.00 (0.00)	NA
GQS	2.74 (0.63)	2.78 (0.65)	2.63 (0.58)	0.288

## Data Availability

Data are available upon request.

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
