# Peer review of "How Useful Is TikTok for Patients Searching for Carpal Tunnel Syndrome-Related Treatment Exercises?"

_healthcare, 2024, doi:10.3390/healthcare12171697_

Round 1

Reviewer 1 Report

Comments and Suggestions for Authors

There are generally no papers on this topic, or a small number of papers have been published that include other social networks (mainly YouTube) or other health condition. Therefore, this topic and this research results are potentionaly interesting for the readers.

However, I have a few comments.

1) First, the introduction does not mention the results and findings of similar studies that have used other platforms (YouTube), yet the authors build on these results in the discussion. This previous research should be mentiond in introduction.

2) In Figure 1, it can be seen that 135 videos were marked as duplicates; however, this is not mentioned anywhere in the text. This part should be explained in the methods section.

3) All the scales for evaluating the videos are explained in detail, and it is explained how they are conducted. Only for the JAMA benchmark criteria this is not the case. This scoring method should be explained in more detail.

4) In the first part of the results, the search method is described, stating how many videos were initially obtained and how many were excluded. This information does not belong in the results section but in the methods section, and if it is already mentioned there, there is no need to repeat it.

5) The information whether whether the evaluation was blinded is missing. Did the researchers who scored the videos know who posted the video, i.e., whether a particular video was created by a healthcare professional or a general user? If they knew, does this represent a certain bias? It should definitely be emphasized whether they were aware of this information when evaluating the videos.

6) It would be clearer if the results for the JAMA and GQS scores were presented in a separate paragraph with a separate subheading, as was done in the methods section when describing the scales.

7) Perhaps a potential explanation could be offered as to why the videos posted by healthcare professionals were rated as poor?

Reviewer 2 Report

Comments and Suggestions for Authors

Overall this is an intereting study. As the authors surveyed a good number of videos, it will add more value if data analysis can go further. 

Lines 183-186: Based on the medians and IQRs, the distributions of those measurements seem quite skewed. It might be interesting to read your explainations in the paper. It will be good to investigate whether there is significant differences among vidoes uploaded by professionals in terms of effectiveness. 

Lines 263- 279 can be placed in the background section of the paper. They draw nothing from the study results.

Applying subheadings in the Discussion section may help highlight the essential findings the authors wish to claim in this study

Reviewer 3 Report

Comments and Suggestions for Authors

1.  The novelty/ significance of the study is not evidence, what is the motivation to perform this research? It is known that social media, for example, TikTok is meant to distribute info mostly for entertainment purposes. The content may or may not be moderated or reviewed, as long as the content does not violate normal social norms. With that, why should one think using TikTok to source medical-related info is reliable? 

2. Line 58, please provide references to support the evidence. For example, a study on health care information on the TikTok platform for Covid-19, etc.

3. Explain why and how the proposed search strings are established. The methodology shall be further detailed to improve clarity. At least the methodology should be structured and technically sound.
